# Meandering Characteristics of the Yimin River in Hulun Buir Grassland, Inner Mongolia, China

**Yuanyuan Zhou** [1] **and Qiuhong Tang** [1,2,*]

1. Key Laboratory of Water Cycle and Related Land Surface Processes, Institute of Geographic Sciences and Natural Resources Research, Chinese Academy of Sciences, Beijing 100101, China; zhouyy.09b@igsnrr.ac.cn
2. University of Chinese Academy of Sciences, Beijing 100049, China
* Correspondence: tangqh@igsnrr.ac.cn

**Abstract:** The evolution of meandering rivers continues to attract considerable attention in research and for practical applications, given that it is closely associated with the safety of river systems and riparian zones. There has been much discussion regarding the various channel planform features exhibited by meandering rivers under different river systems and riparian conditions. The Yimin River is a good example and is located southeast of the Hulun Buir Grassland, which is characterised by a fragile ecosystem and little anthropological activity along with active flow during the non-frozen season from May to November each year and relatively low sediment discharge compared with the Yellow River and Mississippi River. Improved analysis of the evolution of the Yimin River from 1975 to 2019 can support increased local species diversity and more effective flood risk and river management. With the combined Google Earth Engine (GEE) platform and the Geographic Information Systems (GIS) technique, remote sensing images, including Landsat images and global surface water data, are used to analyse the channel planform features of the freely meandering river channel in the middle and lower Yimin River. The results show that the percentage of low sinuosity channel bends was higher than that of high-sinuosity bends. Although the bends with an amplitude greater than 0.48 km and sinuosity greater than 2.3 have an evident upstream-skewed trend, the main channel planform features were downstream skewed with 1499 such bends. The river system conditions in the Yimin River, including lower sediment discharge and vegetation cover, are conducive to the development of downstream-skewed bends. The high-sinuosity bends were found to have a relatively larger ratio during 1981–2000, a period with higher mean annual streamflow compared with other time periods.

**Keywords:** meandering river; Yimin river; Google Earth Engine; planform features; skewed direction





## 1. Introduction

The word "meander" comes from the Büyük Menderes River, which originates in southwestern Turkey [1–4]. River meandering is a common natural geomorphic feature and is also one of the most dynamic [5,6]. It is seen widely in alluvial plains, grasslands, tropical rain forests, and desert margins [7,8]. Meandering rivers exhibit migration of the channel location in their floodplains with changing sinuosity due to outer bank erosion and inner bank deposition. The evolution of meandering rivers accompanied by river erosion and sedimentation has a profound effect on the exploitation and preservation of river systems and riparian zones, including flood alleviation, channel maintenance, riverbank erosion, land protection, and infrastructure resilience [9]. Therefore, it has long attracted the interest of scientists [10–15].

The development of a meandering pattern results from bend instability due to interactions between flow, sediment transport, and bank erosion. In general, due to the channel morphological characteristics and flow momentum, water and sediments converge in the concave banks and diverge in the convex banks. With the flow momentum redistribution,

riverbeds in concave banks gradually degrade and deposit near convex banks until point bars form [16]. Meandering loops constantly migrating in a downstream direction cause the evolution of meanders in rivers. During this evolution, bends display several morphological features which can be captured clearly in high-sinuosity and high-amplitude stages; in natural meandering rivers, these are primarily asymmetric and display upstream skewness and downstream skewness [17,18]. Therefore, skewed meander forms are considered typical of meandering rivers [19,20].

The evolution of meandering bends is usually characterised by geometric parameter adjustments, such as meander wavelength, neck–mouth width, river width, curved top width, meander axis length, bend deflection angle, bend radius, channel centreline curve length, and sinuosity [21,22]. In the current literature, the DEM data and GIS technologies, remote sensing technology, and mathematical modelling were the common methods for obtaining geometric parameters. For example, Bag et al. [23] assessed the meander geometry changes of the Bhagirathi River using remote sensing and GIS techniques and found that the river channel shows an unstable behaviour due to the higher rate of migration of meandering bends. Using remote sensing images, Yousefi [8] analysed the influence of land-use change on the evolution of the Karoon River, Iran. Guo et al. [14] simulated the evolution process of 20 reaches of freely meandering alluvial rivers using the Kinoshita curve. The complex planform results from different environmental factors, including hydrodynamic characteristics, bed material characteristics, soil properties, land use and land cover, and floodplain topography [24–28]. Abad and Garcia [25] suggested that hydrodynamics was the main influential factor at short time scales, whereas at intermediate time scales and longer time scales, bed evolution and bank retreat were the main controlling factors.

Several researchers have proposed that an upstream-skewed planform was a feature of natural meandering streams [17,19,29,30]. For example, Marani et al. [30] found that consistent upstream skewing was the main platform feature of the river Livenza in northern Italy. Vermeulen et al. [31] found that the temperate Red River and the tropical Purus and Mahakam Rivers showed evident upstream skewness. However, Vermeulen et al. [28] found that there was a slight difference between upstream skewed and downstream skewed in the tropical Kapuas River. Fernández and Parker [32] found that meltwater channels with lower sinuosity values had more downstream-skewed bends in Iceland and Canada and proposed that sediment deposition contributed to the development of upstream skewness, and downstream-skewed bends were more common in environments with an absence of sediment deposition.

Researchers have also carried out a series of experimental investigations and tested models for the thermal and hydrodynamic considerations of planimetric instabilities. Güneralp and Rhoads [27,33] found that upstream and downstream skewness were generated when the patch sizes of landscapes were larger and smaller, respectively, than the initial meander size. Seminara et al. [34], Lanzoni and Seminara [35] proposed that bends skewed upstream for the sub-resonant conditions and downstream for the super-resonant conditions based on the hypothesis of a direct relationship between planimetric instabilities and bend skewness. Zolezzi and Seminara [36], and Frascati and Lanzoni [37] also found this phenomenon. Abad and Garcia [26] found the erosional area was mostly concentrated along the outer bank for the upstream-skewed condition, whilst the downstream configuration displayed a higher rate of scouring power and migration rate and larger height and wavelength. However, other influential factors, which were not accounted for in the models and experiments in real rivers, caused different planform features. Perucca et al. [38] described that downstream skewness could develop in the sub-resonant meandering rivers due to the influence of riparian vegetation.

Compared with the conditional methods, remote sensing data contain abundant information in continuous space and time that has been widely used to discuss the river channel change. When detecting the channel planform changes using remote sensing images, the determination of the river channel boundary is the critical step. The boundary of the water surface was usually defined as the channel boundary [39]. However, the

water level change at different times would result in errors in the process of detecting river channel dynamics using remote sensing images. To reduce the error, the riparian vegetation and revetment projects were used to define the boundary [40]. In recent years, remote sensing images at the same acquired date every year in the study period were widely used to detect the river channel boundary [40,41]. In addition, Google Earth Engine (GEE) is a cloud-based platform that processes large amounts of freely available satellite imagery online. In recent years, researchers attempted to apply GEE to analyse the channel changes for its efficient computing power. It is the key step for extracting the water surface change detection indexes by GEE, such as NDVI, NDWI, and MNDWI [42–44]. Based on the water surface change indexes, morphological indicators are used to assess the channel dynamics. Although relatively little literature were found on the river evolution area by GEE, the results show that GEE has a better performance [45]. For example, Swe et al. [46] found that the channel pattern between Pyay to Hithada segment was transitional between straight and regular from 1991 to 2015. Rahman [47] found that it showed a gradual decrease and then a sharp increase in the braided area of the Brahmaputra River. Boothroyd et al. [48] found that the average migration rate was 17.5 m/yr from the Bislak and Cagayan Rivers in the Philippines during 1988–2005.

Although substantive progress has been made from experiments, model simulations, and remote sensing extraction, research conclusions are inconsistent due to the different environmental conditions of meandering rivers. Therefore, further research is required to ascertain the planform regularity of meandering rivers in a variety of environments [3], especially freely meandering rivers. The Yimin River is located in the Hulun Buir Grassland, which is a semi-arid meadow steppe. Due to limited human activities to protect its fragile ecology, the river system of the Yimin River is generally only minimally influenced by humans. The vegetation of this region is abundant, and a variety of rare plants and animals live here, such as Pinus sylvestris var. mongolica, Salix hsinganica, Great bustard, Red-crowned cranes, and Whooper swans. This river flows actively from May to November every year with relatively low sediment discharge. In this study, based on the extracted water surface using the NDWI index from the remote sensing images by the use of GEE and ArcGIS, the meander geometry parameters were estimated for 1975, 1980, 1985, 1990, 1995, 2000, 2005, 2010, 2015, and 2019 to assess the planform features of the Yimin River, and hopes to further inform discussion on the evolution of characteristics under the different environmental conditions of the freely meandering rivers. Moreover, the conclusions could also help to maintain local species diversity and the safety of people and properties in the riparian and downstream regions.

## 2. Materials and Methods

### 2.1. Study Area

The Yimin River is a tributary of the Hailar River. It originates from the northern slopes of Mushroom Mountain in the southeast of Honghuarji Town, Ewenki Autonomous Banner. This river flows from south to north through the Honghuarki Town and Bayan-tohai Town and runs into the Hailar River in the Hailar District (Figure 1). The total length is approximately 360 km, with a basin area of approximately 22,640 km$^2$. The river width is approximately 20–50 m upstream and 50–80 m in the middle and downstream. The Honghuaerji hydrological station is the boundary between the upstream and midstream. Its tributaries include the Weina River, Weizikeng River, Sini River, and Hui River. The Hailar station is the last hydrological gauging station in the lower Yimin River.

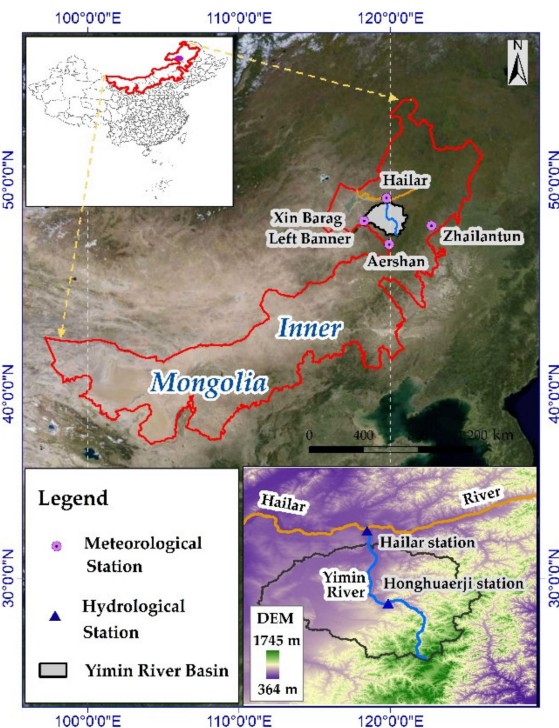

**Figure 1.** The location of study area, hydrological stations, and meteorological stations in and around the study area.

The terrain of this region consists of a basin, hills, and mountains, with an average elevation of 600–900 m. The mountainous land covered with woodland is located in the upper reaches. Hills and basins covered with grass are located in the middle and lower reaches.

This area has a mid-temperate continental monsoon climate with mild, wet summers and cold, dry winters. According to the meteorological stations of Xin Barag Left Banner, Hailar, Aershan, and Zhalan Tun in the period from 1960 to 2020, the annual mean precipitation of the Yimin River basin was 368 mm, annual mean temperature, mean maximum temperature, and the mean minimum temperatures are 1.2 °C, 5.5 °C, and −7.2 °C, respectively. In winter, the river is frozen during the lower temperatures from December to the early April of next year. According to the observed runoff and sediment data in the Hailar hydrological station, the average annual runoff was $10.5 \times 10^8$ m$^3$, and the average annual sediment concentration and sediment discharge were 0.15 kg/m$^3$ and $17.4 \times 10^4$ t, respectively [49,50].

The soil in the riverbank is silty sandy loam with the character of sufficiently loose and poor impact resistance. According to the land-use data obtained from the Resource and Environment Data Cloud platform of the Chinese Academy of Sciences (http://www.resdc.cn, accessed on 15 March 2022), the area changes of land-use type within a 10 km buffer (5 km from the left bank and 5 km from the right bank) around the channel centreline are shown in Table 1. The percentage of area changes was no more than 6% in 1980–1990, 1990–2000, 2000–2010, 2010–2020, and 1980–2020. Between 1980 and 2020, the area of forest and grassland decreased by 5.5%, and the cropland and urban area increased by about 3%.

**Table 1.** The area changes of land use type in 1980 and 2020.

| | 1980 | 1980–1990 | 1990 | 1990–2000 | 2000 |
|---|---|---|---|---|---|
| Land-use type | Area (km$^2$) | Percentage of changes (%) | Area (km$^2$) | Percentage of changes (%) | Area (km$^2$) |
| cropland | 20 | 0.1 | 21 | 1.6 | 33 |
| forest and grassland | 457 | −0.1 | 456 | −2.7 | 435 |
| urban area | 10 | 1.0 | 18 | −0.1 | 17 |
| bare area | 283 | −0.8 | 277 | 1.0 | 285 |
| | 2000–2010 | 2010 | 2010–2020 | 2020 | 1980–2020 |
| Land use type | Percentage of changes (%) | Area (km$^2$) | Percentage of changes (%) | Area (km$^2$) | Percentage of changes (%) |
| cropland | 1.4 | 44 | 0.1 | 45 | 3.2 |
| forest and grassland | −0.8 | 429 | −1.8 | 415 | −5.5 |
| urban area | 0 | 17 | 2.1 | 33 | 3 |
| bare area | −0.6 | 280 | 0 | 280 | −0.4 |

*2.2. Dataset and Methods*

2.2.1. Remote Sensing Images

The yearly global surface water (GSW) dataset from 1984 to 2020 is available in the GEE [51]. The data are generated from Landsat 5, 7, and 8 between 1984 and 2020, and each pixel is individually classified into seasonal water/permanent water/non-water. The surface water distribution of the Yimin River was extracted from GSW in 1985, 1990, 1995, 2000, 2005, 2010, 2015, and 2019 by use of GEE. Considering the influence of the water level change, each pixel of the selected images is classified into either water/no water before extracting the water boundary. However, it does not include the data from before 1984. Therefore, the normalised difference water index (NDWI) from Landsat 2–3 was used to extract the surface water data between 1975 and 1980 with a cloud cover of less than 50%. The mean annual NDWI was used for the analysis of the water surface variation through calculating the average NDWI of all of the remote sensing in 1975 and 1980.

The NDWI were calculated using green and NIR wavelengths:

$$NDWI = (\rho green - \rho NIR)/(\rho green + \rho NIR) \tag{1}$$

where $\rho$green and $\rho$NIR are the radiances of the green and near-infrared wavelengths. To reduce the error as much as possible, the average NDWI of all of the remote sensing between 1975 and 1980 were calculated for the Yimin River. The ArcScan tool in ArcGIS 10.6 was used to extract the centreline of the water surface as the river channel centreline [39].

The normalised difference vegetation index (NDVI) is often used to detect vegetation dynamics. To analyse the influence of riparian vegetation in the evolution of the Yimin River, the mean annual NDVI was extracted by use of GEE. The cloud cover in selected images were less than 50% in 1975, 1980, 1985, 1990, 1995, 2000, 2005, 2010, 2015, and 2019. The mean annual NDVI were obtained by calculating the average NDVI of all of the remote sensing in the study years.

The NDVI was calculated as follows:

$$NDVI = (\rho NIR - \rho red)/(\rho NIR + \rho red) \tag{2}$$

where $\rho$red and $\rho$NIR are the radiances of the red and near infrared wavelengths.

Based on the extracted channel centreline, simple bends and planform geometric parameters were assessed to analyse the spatial features of rivers [43]. The NDVI inner bends were extracted in 1975, 1980, 1985, 1990, 1995, 2000, 2005, 2010, 2015, and 2019 to analyse the relationship between vegetation cover and the channel planform features. However, the NDVI of inner bends were negative in 1975 and 1980. Thus, the extracted

NDVI were ignored in 1975 and 1980, and the relationships between the NDVI of the inner bends and the planform geometric parameters were assessed in 1985, 1990, 1995, 2000, 2005, 2010, 2015, and 2019.

### 2.2.2. Channel Planform Geometric Parameters

The geometry of meandering channels can describe the spatial feature of rivers [52] (Figure 2).

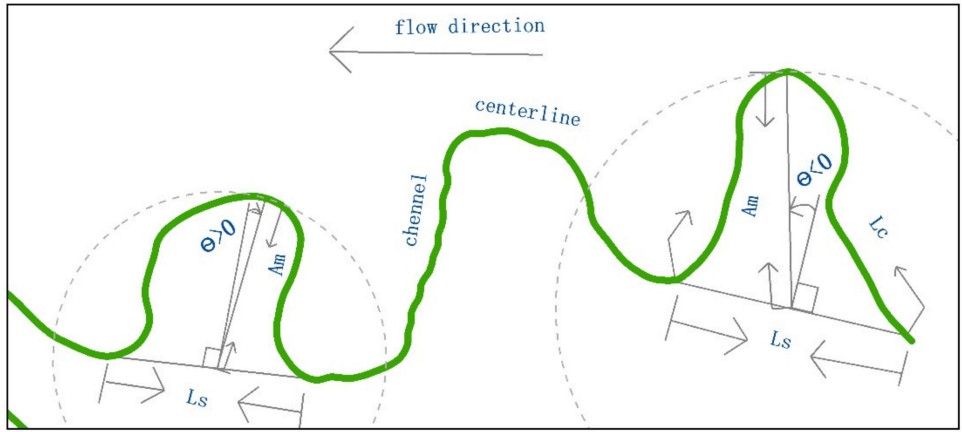

**Figure 2.** Channel planform geometric parameters.

Meander Amplitude (Am) is the maximum distance from the down-valley axis to the sinuous axis of a meander loop. Wavelength (Ls) is the length between the troughs or crests of a meander. Direction angle ($\Theta$) shows the angle between the path and the down-valley direction.

Based on the tool of the line turns to points in ArcGIS, the segment of Ls and Am were manually obtained through the identification of the peak and valley positions from the group points. The length of Am, Ls, and Lc and the direction angle ($\Theta$) were automatically measured by the ArcGIS tool.

The sinuosity index is the ratio of channel length to valley length, which is commonly used to calculate sinuosity changes in the river channel. Sinuosity index (C) was calculated using the following equation:

$$C = Lc/Ls \tag{3}$$

where Lc is channel centreline length, and Ls is water flow length or wavelength. The channels type can be classified into three classes: straight channels (C < 1.05), sinuous channels (C = 1.05–1.5), and meandering channels (C > 1.5) [53].

## 3. Results

Based on the channel centrelines in 1975, 1980, 1985, 1990, 1995, 2000, 2005, 2010, 2015, and 2019 (Figure 3), the geometric parameters of the simple bends were calculated. The total number of channel bends was 1499. In 1975, 1980, 1985, 1990, 1995, 2000, 2005, 2010, 2015, and 2019, the number of bends were 149, 138, 134, 167, 151, 151, 167, 144, 144, and 154, respectively.

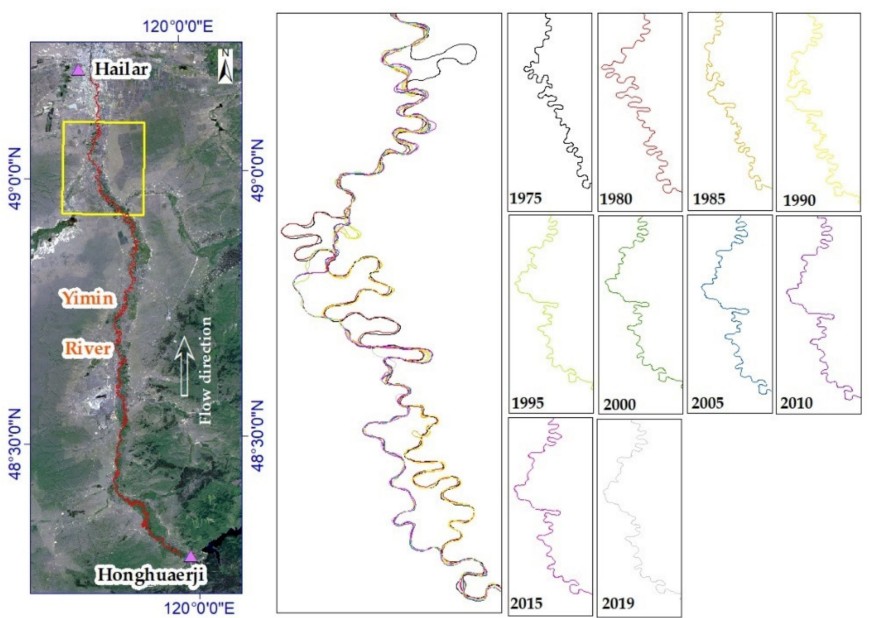

**Figure 3.** The extracted channel centreline in 1975, 1980, 1985, 1990, 1995, 2000, 2005, 2010, 2015, and 2019.

### 3.1. Channel Geometric Parameters

The measured meander wavelengths, amplitudes, and direction angles are exhibited in Figures 4–6.

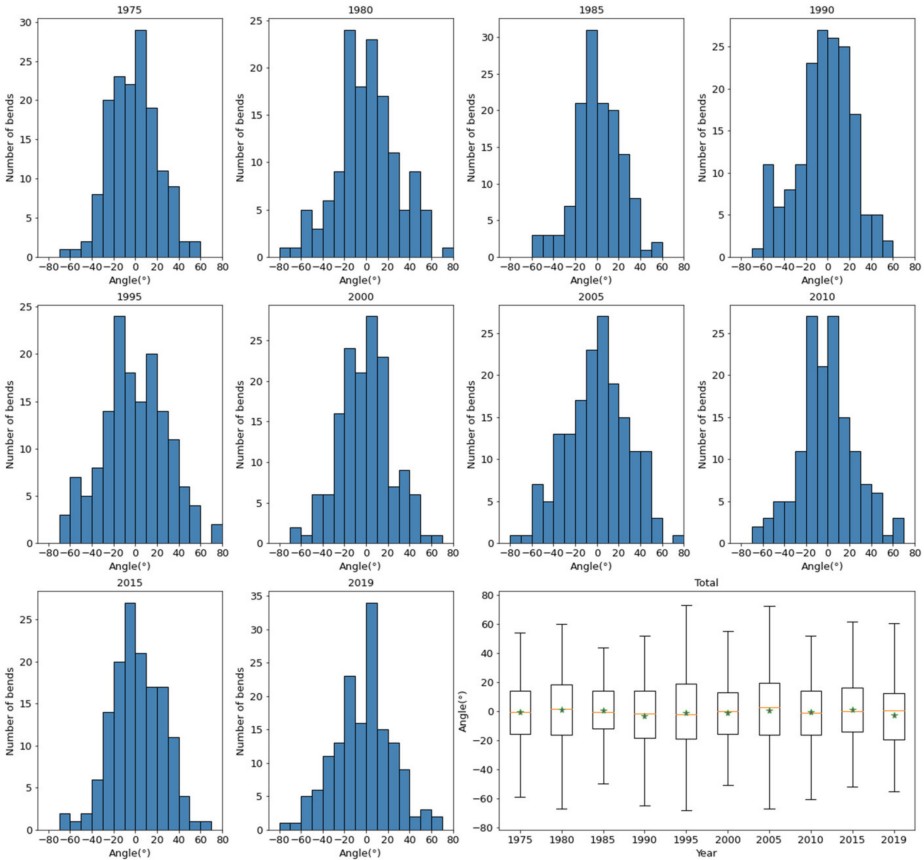

**Figure 4.** The measured meander wavelength in 1975, 1980, 1985, 1990, 1995, 2000, 2005, 2010, 2015, and 2019.

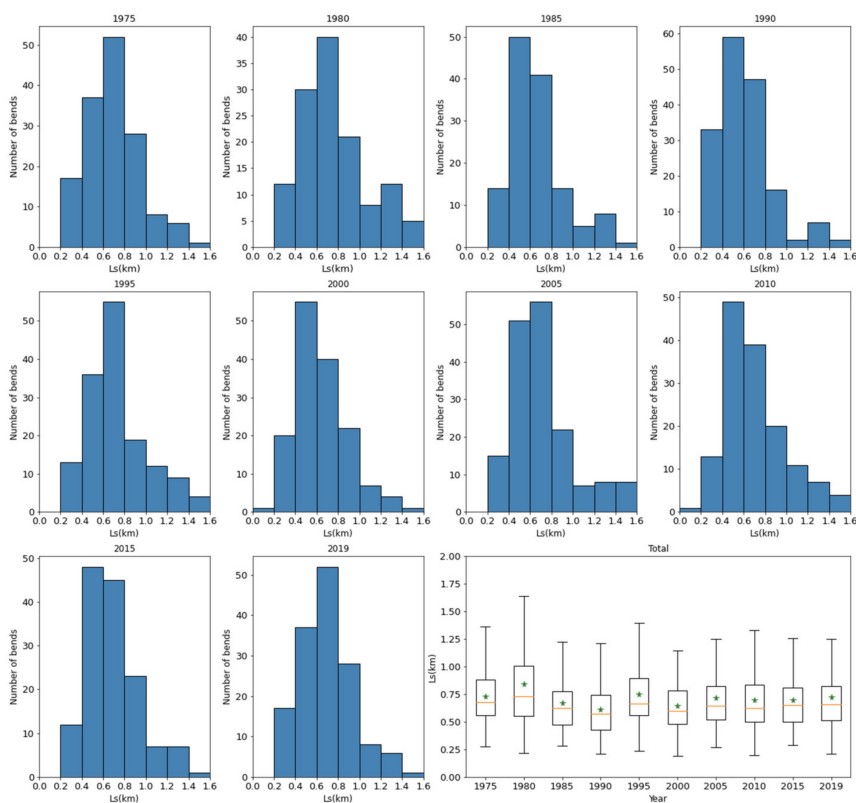

**Figure 5.** The measured meander amplitude in 1975, 1980, 1985, 1990, 1995, 2000, 2005, 2010, 2015, and 2019.

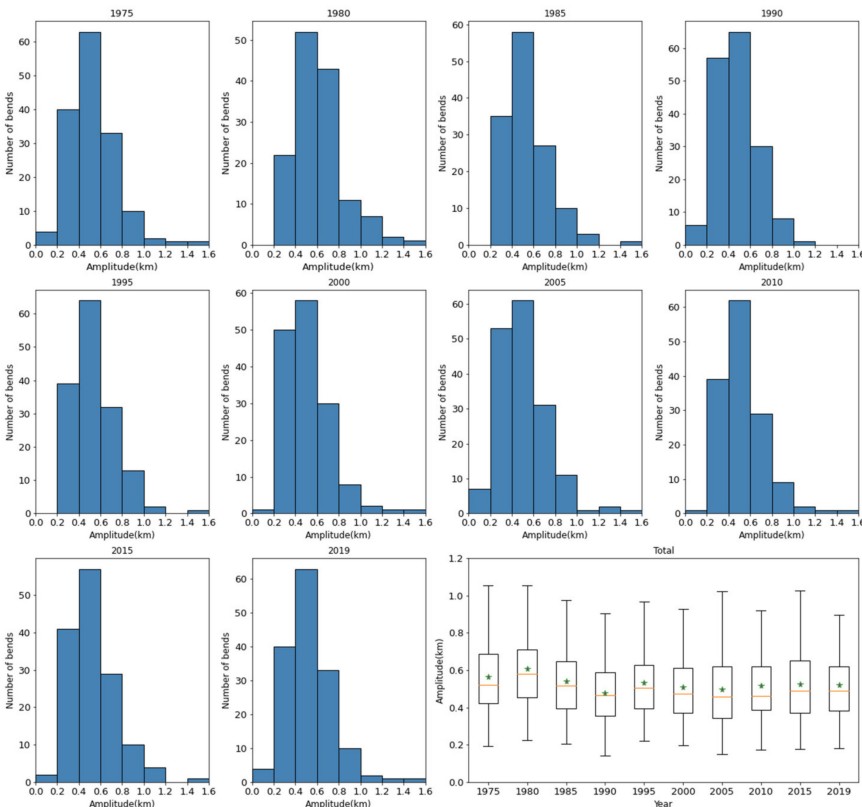

**Figure 6.** The measured meander direction angle in 1975, 1980, 1985, 1990, 1995, 2000, 2005, 2010, 2015, and 2019.

Figure 4 shows that the direction angle (Θ) range of the 1499 bends was between −75°
and 76°. The highest frequency angles ranging from −20° to −10° were 17.4% in 1980
and 15.9% in 1995. The highest frequency angles ranging from −10° to 0 were 23.1% in
1985, 16.2% in 1990, and 18.9% in 2015. The highest frequency angle ranging from 0 to
10° accounted for 19.5%, 18.5%, 16.2%, 18.8%, and 22.1% in 1975, 2000, 2005, 2010, and
2019, respectively.

Figure 5 shows that the wavelength (Ls) range was between 0.18 km and 2.33 km,
with a mean wavelength of 0.71 km. Overall, the highest percentage wavelength range was
between 0.4 and 0.8. The highest frequency wavelengths ranged from 0.6 km to 0.8 km
and accounted for 39.6%, 29%, 36.4%, 30.5%, and 33.8% in 1975, 1980, 1995, 2005, and 2019,
respectively. The highest frequency wavelength ranging from 0.4 km to 0.6 km accounted
for 37.3%, 35.3%, 26.5%, 34%, and 33.3% in 1985, 1990, 2000, 2010, and 2015, respectively.

Figure 6 shows that the amplitude of meanders ranged between 0.14 km and 1.55 km,
and the mean amplitude was 0.53 km. The highest frequency amplitude was between
0.4 km and 0.5 km, with percentages of 18.8–23.5%.

### 3.2. Skewing Direction and Sinuosity Index

The sinuosity range of 1499 bends were between 1–7 (Figure 7).

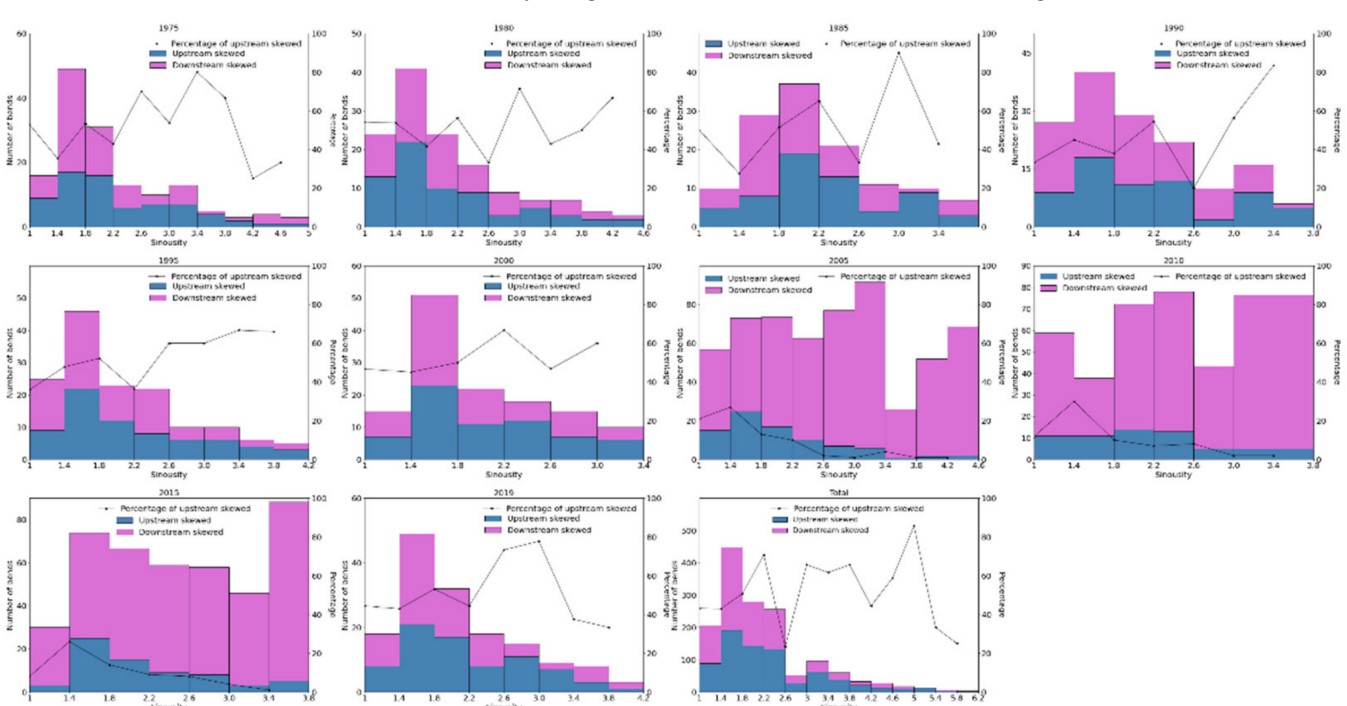

**Figure 7.** The distribution of sinuosity and skewing angle of all the 1499 bends.

Figure 7 shows that low-sinuosity channel bends occurred at a relatively higher
percentage, and high-sinuosity bends occurred at a relatively lower percentage. The per-
centages of low sinuosity with a sinuosity range of 1–2.2 accounted for over 50% of the total
number of bends, and the percentages were 63.8%, 64.5%, 57%, 57%, 62%, 58%, 71%, 61%,
63%, and 64% in 1975, 1980, 1985, 1990, 1995, 2000, 2005, 2010, 2015, and 2019, respectively.
In the period of 1980–2000, the percentages of low sinuosity were relatively small. The
highest percentage of sinuosity with a sinuosity ranging from 1.4 to 1.8, accounted for 32%,
30%, 24%, 30%, 34%, 31%, 28%, 35%, and 32% in 1975, 1980, 1990, 1995, 2000, 2005, 2010,
2015, and 2019, respectively. However, the highest percentage of sinuosity with sinuosity
ranging from 1.8 to 2.0 accounted for 28% in 1985.

The number of upstream-skewed bends and downstream-skewed bends showed that
the bends were primarily downstream-skewed. Viewed in total, 50.4% of the bends were
downstream skewed, and 49.6% were upstream skewed. The percentage of upstream

skewed accounted for 52%, 49%, 51%, 52%, 52%, 50%, 48%, 51%, 50%, and 49% in 1975, 1980, 1985, 1990, 1995, 2000, 2005, 2010, 2015, and 2019, respectively.

### *3.3. Skewing Direction and Sinuosity Index in High-Amplitude Bends*

To assess the distribution of the skewing angles under different amplitudes, the quartile method was used. Figure 8a shows the mean and median skewing angles as a function of amplitude. It shows that equal frequencies of upstream and downstream-skewed bends occurred with amplitudes of approximately 0.48 km. The number of upstream skewed with amplitude greater than 0.48 km accounted for 53% of the total number of bends. Therefore, the bends with an amplitude greater than 0.48 km have an evident trend of upstream skewness.

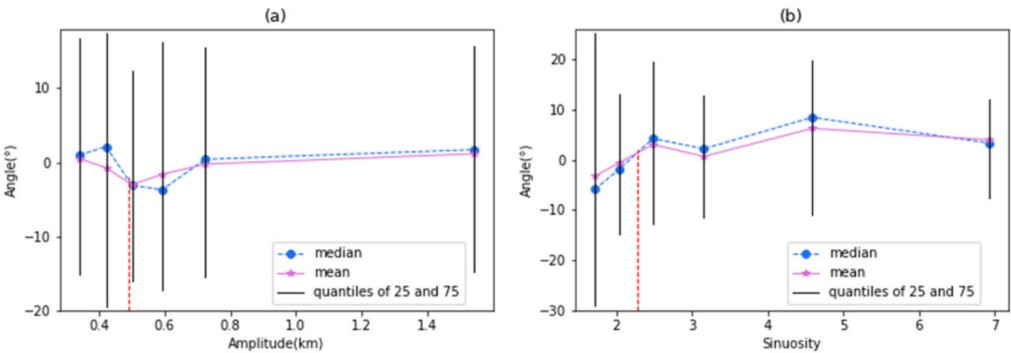

**Figure 8.** The skewing angle changes with amplitude and sinuosity. (**a**) the skewing angle changes with different amplitude, (**b**) the distribution of skewing direction under different sinuosity.

Based on the assessment of skewing angle variation with amplitude, the bends with an amplitude greater than 0.48 km as high amplitude were used to analyse the skewing direction change with increasing sinuosity. Quartiles were also used to assess the distribution of skewing direction under different sinuosities. Figure 8b shows the mean and median skewing angles as a function of sinuosity. With a sinuosity of approximately 2.3, upstream and downstream-skewed bends occurred with equal frequency. The bends with a sinuosity greater than 2.3 were predominantly upstream skewed.

## 4. Discussion

The planform geometry of the Yimin River shows a prominent characteristic of low-sinuosity bends occurring at a relatively higher percentage and high-sinuosity bends occurring at a relatively low percentage, which is a common feature in freely meandering rivers. The phenomenon has also been observed in other meandering rivers, such as upland rivers in the western North Pacific [54], approximately 10,000 tidal and fluvial meanders worldwide [55], and other meandering rivers worldwide [6]. The process of meandering river evolution could explain this phenomenon. Meandering river evolution is the developmental process of meandering loops from low sinuosity to high sinuosity and cut-off. Due to the channel planform and inertial forces, at the early stage of low sinuosity, meandering loop migrations tend towards the downstream direction. With the increasing sinuosity, meandering loops reduce downstream migration and primarily extend laterally until reaching a quasi-stable state. When sinuosity continues to increase, meandering loops migrate towards the upstream direction and extend laterally until a cut-off occurs [16]. Therefore, high sinuosity means migration and cut-off of the active meandering river. This may be the reason that the percentage of low-sinuosity bends is higher than high-sinuosity bends.

During the evolution of meandering rivers, channels exhibit a complex planform. For the nonlinear relationships among river system factors, the bends which are formed are usually asymmetrical. In natural rivers, upstream-skewed forms are considered to be the common planform. However, in the Yimin River, the percentage of downstream skewness

is slightly higher than upstream skewness. The evolution of meandering rivers is influenced by spatial variability in the erosional resistance of floodplain environments [24,56], such as lower versus higher vegetation cover and sediment [6,38,56,57]. Environments where there is an absence of sediment erosion and deposition are conductive to downstream-skewed development [32]. The mean annual sediment discharge of the Yimin River is $17.4 \times 10^4$ t/yr [49]. Compared with the Mississippi River ($1.45 \times 10^8 - 4 \times 10^8$ t/yr) [52], the sediment discharge is relatively low. In the Mississippi River, the bends of upstream and downstream skewness are 60% and 40%, respectively [16,58]. Therefore, the deposition environment of the Yimin River is suitable for the development of downstream skewness.

Moreover, riparian vegetation is an important factor in the evolution of meandering rivers [59]. To establish the relationship between vegetation cover changes and skewed direction in the Yimin River, the distribution of skewing angle changes with increasing NDVI was calculated (Figure 9).

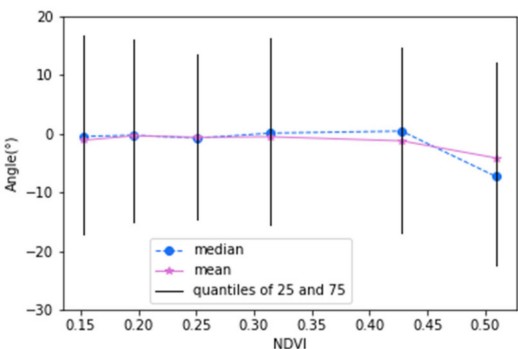

**Figure 9.** The skewing angle changes with vegetation cover in Yimin river.

Figure 9 shows that bends have an increasing trend of downstream skewing with increasing vegetation cover in the Yimin River. Riparian vegetation can change the channel morphology through structural effects that augment flow resistance, cause changes to the flow field and its turbulence structure, reduce the evolution rate of the meandering river, and promote lateral extensions [60–62]. Moreover, several researchers found that the increased flow resistance is more suitable for the development of downstream skewness [36,37,63]. As shown in Figure 9, the Yimin River has the same morphological feature. In other words, the condition of vegetation is an important factor for downstream-skewed bend development in the Yimin River.

Moreover, runoff is an important factor in the process of channel evolution. Runoff, as a kinetic energy source, can promote meander stretch and migration, especially when the runoff is above the average value [64]. In the period of 1980–2000, the high-sinuosity bends had a relatively larger ratio. During 1981–2000, the mean annual runoff of the Yimin River was $14.3 \times 10^8$ m$^3$/yr, which was larger than the mean annual runoff in the periods of 1971–1980 ($9.4 \times 10^8$ m$^3$/yr), 2001–2012 ($6.2 \times 10^8$ m$^3$/yr), and 1970–2012 ($10.5 \times 10^8$ m$^3$/yr) [65]. The relative larger runoff in the period of 1981–2000 might have caused the higher sinuosity bend formation.

## 5. Conclusions

The GEE platform provided an opportunity to perform a multi-temporal analysis of the morphological planform change using Landsat images due to its powerful processing capability. Based on the Landsat images provided by USGS, and the yearly GSW dataset, the water surface and NDVI for the Yimin River in the study years were extracted using the GEE. The channel centrelines were extracted by ArcScan in ArcGIS. A total of 1499 bends were selected to analyse the channel evolution. The channel planform geometric parameters, including direction angle, wavelength, amplitude, and sinuosity, were calculated. The direction angle of the 1499 bends varied between −75° and 76°, and the highest percentage of direction angle was between −20° to 10°. The range of wavelengths was

from 0.18 km to 2.33 km with a mean wavelength of 0.71 km, and the highest frequency wavelength ranged from 0.4 km to 0.8 km. The amplitude of meander ranging was between 0.14 km and 1.55 km, the mean amplitude was 0.53 km, and the highest frequency amplitude was between 0.4 km and 0.5 km. Low-sinuosity channel bends occurred at a relatively higher percentage, and high-sinuosity bends occurred at a relatively lower percentage. The evolution of a meandering river with the process of low-sinuosity bend transformation to high sinuosity may explain this phenomenon. The downstream-skewed bends accounted for 50.4% of the bends, and the upstream-skewed bends accounted for 49.6% of bends. The bends with amplitudes greater than 0.48 km and a sinuosity greater than 2.3 have an evident trend of upstream skewness. It can be concluded that the lower sediment discharge and vegetation cover in the Yimin River are conducive to the development of downstream-skewed bends. Moreover, the relatively larger runoff might have caused the higher sinuosity bend formation in the period between 1981 and 2000 compared with the other time periods. The results of this study might help to prepare better management strategies for grassland ecosystem restoration, river ecosystem protection, and the land use of the floodplain.

**Author Contributions:** Y.Z. conceived of the study, designed the study, analysed the data and writing the manuscript. Q.T. designed and revised the manuscript. All authors have read and agreed to the published version of the manuscript.

**Funding:** This work was supported financially by the National Natural Science Foundation of China (Grant No. 41971004, 41790424) and the Newton Advanced Fellowships.

**Acknowledgments:** The authors are grateful to Nigel Wright for improving the quality of this paper. The authors would like to thank the United States Geological Survey (USGS) for providing free Landsat satellite data.

**Conflicts of Interest:** The authors declare no conflict of interest.

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
