# Peer review of "Meandering Characteristics of the Yimin River in Hulun Buir Grassland, Inner Mongolia, China"

_remotesensing, doi:10.3390/rs14112696_

Round 1
Reviewer 1 Report
The research article “Meandering Characteristics of the Yimin River in Hulun Buir Grassland, Inner Mongolia, China” uses remote sensing and GIS to describe the geometric parameters of meander bends in a catchment that has been minimally influenced by anthropogenic activities. The primary data sources are Landsat imagery and the Global Surface Water (GSW) layer, both of which are available within Google Earth Engine (GEE). The study assesses the geometric parameters of 1499 meander bends at repeat intervals between 1975 and 2019 (number of bends ranged from 138 in 1980 to 167 in 1990). Geometric parameters include the bend wavelength, amplitude and direction angle. The authors identify longitudinal differences in geometric parameters (i.e., upstream characteristics differ from downstream characteristics), but over the ~45 year analysis period most bend parameters remain unchanged (i.e., the system has a limited signal of temporal change).
The literature review in the introduction is detailed and well informed; it draws on relevant examples of meander evolution from numerical and laboratory-based studies and includes useful information on remote sensing workflows. The methods are clearly reported and could be reproduced using GEE and ArcGIS. Results provide quantitative detail on the selected geometric parameters and are displayed using appropriate data visualizations. Factors that contribute towards the morphological characteristics are outlined in the discussion (e.g., role of sediment discharge and riparian vegetation).
Main comments:
Landsat 2-3 imagery was used to extend the record length of the GSW dataset (providing additional data from 1975 and 1980). However, the spectral bands of Landsat 2-3 imagery have a coarser spatial resolution (60 m) than the spectral bands of Landsat 5-7-8 imagery (30 m). How important is the discrepancy in spatial resolution between the Landsat 2-3 and GSW products, and how do the authors account for this difference in their analysis?
On skewing direction and sinuosity index (Section 3.2), what is the threshold for defining low- and high-sinuosity bends? Is this threshold informed by the literature, or has it been arbitrarily defined?
Please ensure that the figures are sufficiently high-resolution to resolve the axis labels and text (i.e., uploaded with 300/600 dpi).
There are some errors between numbered in-text citations and numbered references, please check the numerical formatting of in-text citations and references.
Line specific comments:
L14 – the meaning of “active flow” is not clear (i.e., is there no flow before May?) – rephrase to “peak flow” or “peak discharge”.
L15 – meaning of “low” is relative/subjective, can you specify the specific sediment discharge?
L86 – unclear what is meant by conditional in this context – suggest to removing the first part of this sentence.
L91 – add supporting references that use multispectral indices for riverscape characterization e.g.,
Spada, D., Molinari, P., Bertoldi, W., Vitti, A. and Zolezzi, G. (2018). Multi-temporal image analysis for fluvial morphological characterization with application to Albanian Rivers. ISPRS International Journal of Geo-Information, 7(8), 314. https://doi.org/10.3390/ijgi7080314
Boothroyd, RJ., Nones, M. and Guerrero, M., (2021). Deriving planform morphology and vegetation coverage from remote sensing to support river management applications. Frontiers in Environmental Science, 9. https://doi.org/10.3389/fenvs.2021.657354
L95-96 – Error with in-text citation and numbered references: “Swe et al.,” should be [38], “Rahman et al.,” should be [37]. Please check that the numbered citations correspond to the numbered references throughout the manuscript.
L107-108 – could any examples of rare species be added to provide contextual information?
L160-163 – please specify whether a single image was used to calculate the mean annual NDVI, or whether a composite of many images was used (e.g., the average of 20 images with cloud cover < 50% in 2000).
L228-229 – How are low-sinuosity and high-sinuosity bends being categorized? Is there a formal definition to threshold bend types, or is the threshold arbitrarily defined? The rationale for the threshold should be provided in the manuscript.
L244-45 – replace “the method of quartile was used” with “the quartile method was used”.
L286 – units should be t/yr.
L314-315 – this claim is not fully supported by the data – traditional methods are not tested in the manuscript and it is morphological change rather than landform change that is the focus of the investigation. Rephrase to say that “the GEE platform provided an opportunity to perform multi-temporal analysis of morphological planform change” or equivalent.
Author Response
Responce to Reviewer 1 Comments
- Main comments:
Comment: Landsat 2-3 imagery was used to extend the record length of the GSW dataset (providing additional data from 1975 and 1980). However, the spectral bands of Landsat 2-3 imagery have a coarser spatial resolution (60 m) than the spectral bands of Landsat 5-7-8 imagery (30 m). How important is the discrepancy in spatial resolution between the Landsat 2-3 and GSW products, and how do the authors account for this difference in their analysis?
On skewing direction and sinuosity index (Section 3.2), what is the threshold for defining low- and high-sinuosity bends? Is this threshold informed by the literature, or has it been arbitrarily defined?
Please ensure that the figures are sufficiently high-resolution to resolve the axis labels and text (i.e., uploaded with 300/600 dpi).
There are some errors between numbered in-text citations and numbered references, please check the numerical formatting of in-text citations and references.
Response: Roy et al. (2016) found that the sensor differences are small, and Vogelmann et al. (2016) found that data representing natural vegetation from Landsat 5, 7, and 8 were comparable and did not indicate a need for major modification prior to use for long-term monitoring. Thus we did not account for the difference in our analysis.
We didn’t define a threshold to distinguish between low-sinuosity and high-sinuosity bends. Instead, based in the calculated sinuosity indices, the high and low sinuosity bends were arbitrarily identified according to the relative value.
The resolution of the figures uploaded in the submission system were 600dpi.
The order number of citations in-text citations and references have been revised.
- SPECIFIC COMMENTS (keyed to line number)
Comment 1:L14 – the meaning of “active flow” is not clear (i.e., is there no flow before May?) – rephrase to “peak flow” or “peak discharge”.
Response: The hydrologically active flow season is from May to November each year in the Yimin River, because the river is frozen from December to the early April of next year. The peak flow always occurs during July to August each year. So we think “active flow” is better than “peak flow”.
Comment 2:L15 – meaning of “low” is relative/subjective, can you specify the specific sediment discharge?
Response: The average annual sediment concentration and sediment discharge are 0.15 kg/m3 and 17.4×104 t, respectively. This data is mentioned in section 2.1. compared with the Yellow River and Mississippi River, the sediment discharge in Yimin River is relatively lower. The sentence in L15 has been revised into “ …… with active flow during the non-frozen season from May to November each year and relatively low sediment discharge compared with the Yellow River and Mississippi River.”.
Comment 3:L86 – unclear what is meant by conditional in this context – suggest to removing the first part of this sentence.
Response: We accept this suggestion and delete this sentence in manuscript.
Comment 4:L91 – add supporting references that use multispectral indices for riverscape characterization e.g.,
Spada, D., Molinari, P., Bertoldi, W., Vitti, A. and Zolezzi, G. (2018). Multi-temporal image analysis for fluvial morphological characterization with application to Albanian Rivers. ISPRS International Journal of Geo-Information, 7(8), 314. https://doi.org/10.3390/ijgi7080314
Boothroyd, RJ., Nones, M. and Guerrero, M., (2021). Deriving planform morphology and vegetation coverage from remote sensing to support river management applications. Frontiers in Environmental Science, 9. https://doi.org/10.3389/fenvs.2021.657354
Response: We has added references in L91, and this sentence has been revised into “It is the key steps for extracting water surface change detection indexes by GEE, such as NDVI, NDWI and MNDWI [39-41].”. The three references have been added into section “References”.
Comment 5:L95-96 – Error with in-text citation and numbered references: “Swe et al.,” should be [38], “Rahman et al.,” should be [37]. Please check that the numbered citations correspond to the numbered references throughout the manuscript.
Response: We have exchanged the order of reference 37 and 38 in section “References”, and deleted the reference 35. The order number of reference in section 1~4 have been carefully checked and revised in the revision.
Comment 6:L107-108 – could any examples of rare species be added to provide contextual information?
Response: This sentence has been revised into “The vegetation of this region is abundant, and a variety of rare plants and animals live here, such as Pinus sylvestris var. mongolica, Salix hsinganica, Great bustard, Red-crowned cranes, Whooper swans.”.
Comment 7:L160-163 – please specify whether a single image was used to calculate the mean annual NDVI, or whether a composite of many images was used (e.g., the average of 20 images with cloud cover < 50% in 2000).
Response: The mean annual NDVI was extracted though calculating the average NDVI of all the selected remote sensing in study years by use of GEE. This paragraph has been revised into “The normalized difference vegetation index (NDVI) is often used to detect the vegetation dynamics. To analyse the influence of riparian vegetation in the evolution of Yimin River, the mean annual NDVI was extracted by use of GEE. The cloud cover in selected images were less than 50% in 1975, 1980, 1985, 1990, 1995, 2000, 2005, 2010, 2015, 2019. The mean annual NDVI were obtained though calculating the average NDVI of all remote sensing in study years.
”.
Comment 8: L228-229 – How are low-sinuosity and high-sinuosity bends being categorized? Is there a formal definition to threshold bend types, or is the threshold arbitrarily defined? The rationale for the threshold should be provided in the manuscript.
Response: We didn’t define a threshold to distinguish between low-sinuosity and high-sinuosity bends. Instead, based in the calculated sinuosity indices, the high and low sinuosity bends were arbitrarily identified according to the relative value.
Comment 9:L244-45 – replace “the method of quartile was used” with “the quartile method was used”.
Response: This sentence has been revised into “To assess the distribution of skewing angle under different amplitudes, the quartile method was used”.
Comment 10: L286 – units should be t/yr.
Response: The units has been revised into t/yr.
Comment 11:L314-315 – this claim is not fully supported by the data – traditional methods are not tested in the manuscript and it is morphological change rather than landform change that is the focus of the investigation. Rephrase to say that “the GEE platform provided an opportunity to perform multi-temporal analysis of morphological planform change” or equivalent.
Response: this sentence has been revised into “The GEE platform provided an opportunity to perform multi-temporal analysis of morphological planform change using Landsat images due to its powerful processing capability.”.
Reviewer 2 Report
Should be published in current form.
Author Response
Thank you for your review. We have carefully revised this manuscript , and the revised manuscript has been uploaded.
Reviewer 3 Report
This interesting paper focuses on the planform changes of meanders of the Yimin River.
The analysis is based on satellite images and particularly on the water surface extent already available from GEE.
The metrics related to meander geometry were computed considering the water surface. However, the riverbed does not correspond to the water surface only. Several researched defined the active channel. The authors should provide more information to explain what their reference unit for planform analysis is. Considering the “riverbed”, defined as the surface covered by water, namely the water surface extent, is neither enough nor representative, as water level changes over time.
In the introduction a more accurate description should be provided to frame this research. This research focuses on meander geometry, thus more details should be provided about the quantitative analysis of meanders.
The authors try to explain the causes of the planform changes documented. However, no information is reported for example on sediment (grainsize, type), LULC at the catchment scale (and related variations over time), changes in water flux, changes in anthropogenic controlling factors, ... I mean, this work should be more focused on geometrical issues only; actually, no explanation of such dynamics can be reported without an analysis of the reach and catchment scale controlling factors.
Moreover, more details should be provided about the procedures to compute the meander-scale parameters.
Sinuosity is generally computed at the reach scale and based on the active channel (not the wetted channel). Is this river single-thread totally? The physiographical features of this river should be described in detail and the geographical setting map should illustrate more clearly the riverbed.
Finally, it is not very clear how the influence of riparian vegetation on the river evolution was assessed.
Relevant changes are required. In my opinion it is fundamental that the authors focus (i) on the geometrical characterization of meanders (clearly defining what they are analysing – riverbed, active channel, wetted channel?), and (ii) on the explanation of metrics and procedures, (iii) disregarding the causes of such changes and the comparison with other rivers that most probably are characterized by completely different geomorphic and climatic features (different controlling factors).
Some useful references:
Brice, J. C. (1964). Channel patterns and terraces of the Loup Rivers in Nebraska. US Government Printing Office.
Malavoi, J. R., & Bravard, J. P. (2011). Éléments d'hydromorphologie fluviale. Édité par l'Onema (Office national de l'eau et des milieux aquatiques), 2010, 224 p. En ligne sur: http://www. onema. fr/hydromorphologie-fluviale. Physio-Géo. Géographie physique et environnement, (Volume 5), 1.
https://link.springer.com/chapter/10.1007/978-3-030-03515-0_29
https://doi.org/10.1016/S0169-555X(00)00007-6
https://doi.org/10.3390/rs13183775
https://doi.org/10.1002/rra.925
https://doi.org/10.1016/j.geomorph.2022.108280
https://doi.org/10.1016/j.geomorph.2010.09.009
https://doi.org/10.1016/j.catena.2020.105031
https://doi.org/10.1016/j.geomorph.2007.04.034
https://doi.org/10.1016/j.geomorph.2006.06.005
Author Response
Response to Reviewer #3 Comments
Comment1: The metrics related to meander geometry were computed considering the water surface. However, the riverbed does not correspond to the water surface only. Several researched defined the active channel. The authors should provide more information to explain what their reference unit for planform analysis is. Considering the “riverbed”, defined as the surface covered by water, namely the water surface extent, is neither enough nor representative, as water level changes over time.
Response: The boundary of extracted water surface area could be defined as the riverbank line in theory (Yang, C.; Cai, X.; Wang, X.; Yan, R.; Zhang, T.; Zhang, Q.; Lu, X. Remotely Sensed Trajectory Analysis of Channel Migration in Lower Jingjiang Reach during the Period of 1983–2013. Remote Sens. 2015, 7, 16241-16256. https://doi.org/10.3390/rs71215828), and this method has been widely adopted (Jung, H. C.; Hamski, J.; Durand, M.; Alsdorf, D.; Hossain, F.; Lee, H.; Azad Hossain, A.K.M.; Hasan, K.; Khan, A.S.; Hoque, A. Z. Characterization of complex fluvial systems using remote sensing of spatial and temporal water level variations in the Amazon, Congo, and Brahmaputra Rivers. Earth Surf. Process. Landf. 35, 294–304 (2010)). Reducing the influence of water level changes is always a hard problem when define the water surface. The riparian vegetation and remote sensing images at the same acquired date every year were used by researchers. This text in paragraph 6 of section 1 has been revised into “Compared with the conditional methods, remote sensing data contain abundant information in continuous space and time that has been widely used to discuss the river channel change. When detecting the channel planforms changes use the remote sensing images, determination of the river channel boundary is the critical step. The boundary of water surface was usually defined as the channel boundary [36]. However, the water level change in different time would result in the error in the process of detecting river channel dynamic using remote sensing images. To reduce the error, the riparian vegetation and revetment projects were used to define the boundary [37]. In recent years, remote sensing images at the same acquired date every year in study period were widely used to detect the river channel boundary[37, 38]. In addition,……”.
Comment2: In the introduction a more accurate description should be provided to frame this research. This research focuses on meander geometry, thus more details should be provided about the quantitative analysis of meanders.
Response: The last two sentence in the introduction have been revised into “In this study, based on extracted water surface using NDWI index from remote sensing images by use of GEE and ArcGIS, meander geometry parameters were estimated in 1975, 1980, 1985, 1990, 1995, 2000, 2005, 2010, 2015 and 2019 to assess the planform features of the Yimin River, and hopes to further inform discussion on the evolution of characteristics under different environment conditions of freely meandering rivers. Moreover, the conclusions could also help in maintaining local species diversity, the safety of people and properties in riparian and downstream region.”.
We added the literature review about the quantitative analysis of meanders geometry in paragraph 3 section 1, and the added text as fellow:
“The evolution of meandering bends usually descripts by the geometric parameter adjustments, such as meander wavelength, neck mouth width, river width, curved top width, meander axis length, bend deflection Angle, bend radius, channel center line curve length and sinuosity [21, 22]. In the current literature, the DEM data and GIS technologies, remote sensing technology, and mathematical modelling were the common methods obtaining geometric parameters. For example, Bag et al. [23] assessed the meander geometry changes of the Bhagirathi River using the remote sensing and GIS techniques, and found that the river channel shows an unstable behavior due to the higher rate of migration of meandering bends. Using remote sensing images, Yousefi [8] analyzed the influence of land use change to the evolution of the Karoon River, Iran. Guo et al. [14] simulated the evolution process of 20 reaches of freely meandering alluvial rivers using the Kinoshita curve.”.
Comment3: The authors try to explain the causes of the planform changes documented. However, no information is reported for example on sediment (grainsize, type), LULC at the catchment scale (and related variations over time), changes in water flux, changes in anthropogenic controlling factors, ... I mean, this work should be more focused on geometrical issues only; actually, no explanation of such dynamics can be reported without an analysis of the reach and catchment scale controlling factors.
Response: The Yimin River is one of the nature river which has little anthropological activity. The percentage of area changes were not more than 6%. Although the area of cropland and urban area increased from 1980 to 2020, the increased percentage were small compared with the whole study area. Moreover, the runoff change had same trend with rainfall in study area (Bao, G.; Liu, Y.; Liu, N. A tree-ring-based reconstruction of the Yimin River annual runoff in the Hulun Buir region, Inner Mongolia, for the past 135 years. Chinese Science Bulletin, 2012, 57(36), 4765-4775). The average annual sediment concentration is 0.15 kg/m3 which is much lower than Yellow River and Yangze River. Therefore, in this paper, we focus on the morphological features, and this could provide an example for the morphological features of nature meandering river.
The soil feature of river bed and land use type changes have been added into section 2.1. the added text as follow: The soil in the river bed is silty sandy loam with the character of sufficiently loose and poor impact resistance. According to the land use data obtained from the Resource and Environment Data Cloud platform of the Chinese Academy of Sciences (http://www.resdc. cn), the area changes of land use type around 10 km of the channel centerline in 1980 were showed in table1. The percentage of area changes were not more than 6% in 1980-1990, 1990-2000, 2000-2010, 2010-2020 and 1980-2020. During 1980-2020, the area of forest and grassland decreased by 5.5%, the cropland and urban area increased by about 3%.
Table 1 The area changes of land use type in 1980 and 2020
|
  |
1980 |
1980-1990 |
1990 |
1990-2000 |
2000 |
|
Land use type |
Area(km2) |
Percentage of changes (%) |
Area(km2) |
Percentage of changes (%) |
Area(km2) |
|
cropland |
20 |
0.1 |
21 |
1.6 |
33 |
|
forest and grassland |
457 |
-0.1 |
456 |
-2.7 |
435 |
|
urban area |
10 |
1.0 |
18 |
-0.1 |
17 |
|
bare area |
283 |
-0.8 |
277 |
1.0 |
285 |
|
  |
2000-2010 |
2010 |
2010-2020 |
2020 |
1980-2020 |
|
Land use type |
Percentage of changes (%) |
Area(km2) |
Percentage of changes (%) |
Area(km2) |
Percentage of changes (%) |
|
cropland |
1.4 |
44 |
0.1 |
45 |
3.2 |
|
forest and grassland |
-0.8 |
429 |
-1.8 |
415 |
-5.5 |
|
urban area |
0 |
17 |
2.1 |
33 |
3 |
|
bare area |
-0.6 |
280 |
0 |
280 |
-0.4 |
Comment4: Moreover, more details should be provided about the procedures to compute the meander-scale parameters.
Sinuosity is generally computed at the reach scale and based on the active channel (not the wetted channel). Is this river single-thread totally? The physiographical features of this river should be described in detail and the geographical setting map should illustrate more clearly the riverbed.
Response: As figure 2, the meander-scale parameters were obtained based on the extracted river channel centerline. By use of ArcGIS tool, the channel centerline automatic generated from the extracted water surface area in GEE which considered as the river channel area. In Yinmin River, the multithread is rare, and the selected 1499 bends were formed in single thread.
Comment5: Finally, it is not very clear how the influence of riparian vegetation on the river evolution was assessed.
Response: The morphological feature of Yimin River is same with Iceland and Canada channel, and different with Mississippi river. In paragraph 2 of section 4, compared with the Mississippi river, the relatively low sediment concentration may be the main factor for the higher percentage of downstream-skewed bends. Several researchers found that the increased flow resistance is more conducive to the formation of downstream skewness bends (Zolezzi, G.; Seminara, G. Downstream and upstream influence in river meandering. Part 1. General theory and application to overdeepening. Journal of Fluid Mechanics, 2001a, 438(13), 183-211. Perucca, E.; Camporeale, C.; Ridolfi, L. Significance of the riparian vegetation dynamics on meandering river morphodynamics. Water Resources Research, 2007, 43(3)). Figure 9 shows that the Yimin River has the same morphological feature. This sentence has been revised into “Moreover, several researchers found that the increased flow resistance is more suitable for the development of downstream skewness [33,34,60]. As shown in Figure 9, the Yimin River has the same morphological feature. In other words, the condition of vegetation is an important factor for downstream-skewed bend development in the Yimin River.”.
Common6: Relevant changes are required. In my opinion it is fundamental that the authors focus (i) on the geometrical characterization of meanders (clearly defining what they are analysing – riverbed, active channel, wetted channel?), and (ii) on the explanation of metrics and procedures, (iii) disregarding the causes of such changes and the comparison with other rivers that most probably are characterized by completely different geomorphic and climatic features (different controlling factors).
Response: We accept this advice, and the corresponding revised as response to common1~5.
Round 2
Reviewer 3 Report
Dear authors, thank you for the revised version of your manuscript.
Some changes significantly improved the text.
However, I’m still not convinced about the riverbed definition. Certainly, your approach (water surface extraction from satellite images) has been widely used, but the reference polygon obtained for the analysis should be clearly described as water surface and not riverbed in general. Several studies define the riverbed as the active channel (i.e., bars + wetted channel). If this river does not have bars, thus the riverbed could overall be compared to the wetted channel, then you could specify this issue to make clear that the wetted channel you considered can be comparable to the active channel (for example this could happen if the active channel is sinuous or anabranching). This (i.e. the channel type) could be a limit of your analysis that restrict the application possibilities.
References for the riverbed definition:
https://pubs.er.usgs.gov/publication/sir20145112
https://onlinelibrary.wiley.com/doi/abs/10.1002/rra.925
https://www.sciencedirect.com/science/article/pii/S0169555X00000076?via%3Dihub
https://www.sciencedirect.com/science/article/pii/S0169555X13001608
https://www.sciencedirect.com/science/article/pii/S0169555X22001738
https://onlinelibrary.wiley.com/doi/10.1002/esp.328
https://esurf.copernicus.org/articles/8/471/2020/
The sentence “The soil in the river bed is silty sandy loam with the character of sufficiently loose and poor impact resistance” (line 163) is rather strange. No soil is within the riverbed; it is better to refer to bed load; river bed grainsize, or similar.
The new part on LULC refers to a 10-km buffer around the centerline, not to the catchment lulc changes. I believe that you could specify in the text that more accurate analysis could be performed on controlling factors at the reach and catchment scales.
Figure 1: Please add a bare scale in km to make the catchment dimension more clear; please add the unit for elevation (m); please correct the misprint in the third legend entry (Rvier à river)
Figures: in general, the figure quality must be improved as labels are not easily readable.
What is the origin of channel type classification at line 223? Where are the C thresholds from? Could you add a reference?
In section 2.2.2 several parameters are listed. They can be considered to analyse meanders and you explain what they are. However, no information is provided about the procedure for their calculation, namely, the practical way to compute them (in GIS environment? Automatically? Manually?)
Run off (I suggest using the term discharge) values at the end of section 4 should have a unit of measurement (m3).
Author Response
Response to comments
Comment 1: However, I’m still not convinced about the riverbed definition. Certainly, your approach (water surface extraction from satellite images) has been widely used, but the reference polygon obtained for the analysis should be clearly described as water surface and not riverbed in general. Several studies define the riverbed as the active channel (i.e., bars + wetted channel). If this river does not have bars, thus the riverbed could overall be compared to the wetted channel, then you could specify this issue to make clear that the wetted channel you considered can be comparable to the active channel (for example this could happen if the active channel is sinuous or anabranching). This (i.e. the channel type) could be a limit of your analysis that restrict the application possibilities. Response: We agree with your opinion on the definition of riverbed. Our approach extracts water surface to approximate the active riverbed. As this a natural river without bars, the water surface at high wate water level could be a good approximation for the active channel. We have clarified it in the revision. We revised the first paragraph in section 2.2.1, and this text has been revised into “The yearly global surface water (GSW) dataset from 1984 to 2020 is available in the GEE [51]. The data are generated from Landsat 5, 7, and 8 between 1984 and 2020, and each pixel is individually classified into seasonal water/permanent water/non-water. The surface water distribution of the Yimin River was extracted from GSW in 1985, 1990, 1995, 2000, 2005, 2010, 2015, and 2019 by use of GEE. Considered the influence of water level change, each pixel of selected images is classified into water/no water before extracting water boundary. However, ……”. Comment 2: The sentence “The soil in the river bed is silty sandy loam with the character of sufficiently loose and poor impact resistance” (line 163) is rather strange. No soil is within the riverbed; it is better to refer to bed load; river bed grainsize, or similar. Response: this sentence has been revised into “The soil in the river bank is silty sandy loam with the character of sufficiently loose and poor impact resistance.”. Comment 3: The new part on LULC refers to a 10-km buffer around the centerline, not to the catchment lulc changes. I believe that you could specify in the text that more accurate analysis could be performed on controlling factors at the reach and catchment scales. Response: This sentence has been revised into “……, the area changes of land use within 10-km buffer (5-km from left bank and 5-km from right bank) around the channel centerline were showed in Table1.”. Comment 4: Figure 1: Please add a bare scale in km to make the catchment dimension more clear; please add the unit for elevation (m); please correct the misprint in the third legend entry (Rvier à river) Figures: in general, the figure quality must be improved as labels are not easily readable. Response: We have added a scale bar in figure 1, also added the unit for elevation. The word ‘Rvier’ has been revised into ‘River’ in the legend. Comment 5: What is the origin of channel type classification at line 223? Where are the C thresholds from? Could you add a reference? Response: the reference “[53] Das, B.C. Two Indices to Measure the Intensity of Meander. In: Singh, M., Singh, R., Hassan, M. (eds) Landscape Ecology and Water Management. Advances in Geographical and Environmental Sciences. Springer, Tokyo. 2014. https://doi.org/10.1007/978-4-431-54871-3_17. ” has been added into this sentence. Comment 6: In section 2.2.2 several parameters are listed. They can be considered to analyse meanders and you explain what they are. However, no information is provided about the procedure for their calculation, namely, the practical way to compute them (in GIS environment? Automatically? Manually?) Response: The method has been added into paragraph 3 in section 2.2.2, and the added paragraph is as follow: Based on the tool of the line turns to points in ArcGIS, the segment of Ls and Am were manually obtained though identify the peak and valley position from the group points. The length of Am, Ls and Lc, and direction angle (Ѳ) were automatically measured by ArcGIS tool. Comment 7: Run off (I suggest using the term discharge) values at the end of section 4 should have a unit of measurement (m3). Response: We added the unit m3 into the section 4, and this sentence has been revised into “During 1981–2000, the mean annual runoff of the Yimin River was 14.3×108 m3/yr which was larger than the mean annual runoff in the periods of 1971–1980 (9.4×108 m3/yr), 2001–2012 (6.2×108 m3/yr), and 1970–2012 (10.5×108 m3/yr) [65].”.